# Binary-level data dependence analysis of hot execution regions using abstract interpretation at runtime

**Rasha Omar** [1,2]*, **Mostafa Abbas**[1], **Ahmed El-Mahdy**[1,3], **Erven Rohou**[4]

**1** Department of Computer Science and Engineering, Egypt-Japan University of Science and Technology, Alexandria, Egypt, **2** Faculty of Computers and Artificial Intelligence, Benha University, Qalyubia, Egypt, **3** Alexandria University, Alexandria, Egypt, **4** Univ Rennes, Inria, CNRS, IRISA, Rennes, France

* rasha.omar@ejust.edu.eg

**Data Availability Statement:** The article is standalone and we don't need to upload any dataset or supporting files.

**Funding:** The work in this paper is technically funded by the Egyptian Ministry of Higher

## Abstract

With the widespread of multicore systems, automatic parallelization becomes more pronounced, particularly for legacy programs, where the source code is not generally available. An essential operation in any parallelization system is detecting data dependence among parallelization candidate instructions. Conducting dependence analysis at the binary-level is more challenging than that at the source-level due to the much lower semantics of the binary code. In this paper, we consider using the elaborate 'static' analysis of abstract interpretation, for the first time, at runtime for data dependence detection. Specifically, our system interprets instructions at a hot region, while at the same time, collect programs semantics for seen program points, thereby conducting abstract interpretation analysis dynamically. The analysis is guaranteed to be correct as long as execution does not exit the region prematurely. Moreover, successive hot region re-entries will resume previous analysis, albeit much faster in case no major change in the program semantics. Such approach provides for more rigorous analysis than other simple dynamic analysis which would typically miss parallelization opportunities. The proposed approach also does not require any hardware support, availability of the source code, as well as any code re-compilation. To study the performance and accuracy of our approach, we have extended the Padrone dynamic code modification framework, and conduct an initial study on a set of PolyBench kernels and selected programs from SPEC CPU. Experimental results show accurate dependence detection with low overhead.

## Introduction

Recently, multicore architecture is vastly used and accordingly automatic code parallelization becomes a hot concern. Speculative parallelization is a promising technique that can be used to select *potentially* parallel region based on runtime analysis [1], parallelize the code, and provide for rolling-back mechanisms for maintaining correctness, in case of incorrect analysis. Another approach is purely static, where the analysis guarantees that no data dependence exits

Education (MoHE). Also, this work is partially funded by the PHC IMHOTEP project.

**Competing interests:** NO authors have competing interests.

in code regions, which are mainly loops [2]. However, such dependence analysis is a complex task, as it requires proving that all possible execution paths would result in a lack of data dependence. Moreover, the analysis is more complex for the case of binary code, especially for legacy applications and binary code for which the source code is not available. Therefore, the decision of parallelizing of a particular memory word in instruction or data is undecidable. Thus, data dependence in binary-level is challenging [3].

Dependence analysis methods can be broadly categorized into static and dynamic. With static dependence analysis, at compile-time, the compiler conducts conservative analysis to ensure that there is no possible dependence. However, the compiler can incorporate runtime checking for simplifying the static analysis, and decide to execute either serial or parallel accordingly. Nevertheless, once the parallel code executes, correctness is guaranteed.

Dynamic dependence analysis is featured with its for wider applicability especially when dependence violations are rare or when the code complexity is high, thereby making it difficult to prove the lack of dependence. Furthermore, static analysis is generally applicable at the source code level. Since dynamic analysis relies on detecting dependence violations rather than proving their impossibility, it could be applied at binary-level [4].

Various techniques of analysis are used to detect data dependence statically. One of these techniques is Abstract Interpretation (AI), which is an elaborate static analysis method that is generally used in program verification. It simplifies gathering all possible program execution states (collective trace semantics) through abstracting program states and statements. In other words, it provides a set of equations corresponding to each program statement (i.e. the analysis is based on original program semantics). The solution of this equations set generates the program semantics. Abstraction simplifies solving the equations set through a fast converging iterative process. The abstract semantics are a super-set of the actual concrete semantics, provided for sound analysis [5, 6].

In this paper, we consider the case of data dependence analysis of binary code for memory read and write operations using AI. We propose applying AI data dependence technique at runtime for hot binary code regions. In particular, our system identifies the hot function through profiling. Then, the approach interprets each instruction of this function. In doing so, our system could provide corresponding AI equation for each interpreted instruction, and also compute the corresponding current abstract collective semantic value. The interpretation continues until all reachable instructions are executed at least two times with no change in the collected semantics (i.e. reached a fixpoint).

Having the generated semantics, our approach can then compute all possible data dependence through intersecting memory read and write addresses, and generating a corresponding dependence graph. Also, the approach marks all non interpreted region exit instructions, so as they can be used by an underlying speculative execution system to trigger dependence violation and recovering the correct program state program.

To assess the performance and accuracy of our method, we have extended the Padrone binary code modification system to implement the above analysis [7]. The main contributions of this article can be listed as:

- Propose a data dependence analysis technique for the binary code based on AI at runtime;

- Integrate the method into the Padrone binary modification framework;

- Conduct an initial performance investigation on a set of PolyBench kernels to validate the approach.

The remaining part of this paper is organized as follows: Section 1 discusses the related work to our proposed approach. Section 2 provides background on AI, dynamic binary

analysis, and the underlying Padrone binary modification framework, Padrone. Section 3 discusses the main concept of the proposed method. Section 4 provides details about the implementation of our system. Section 5 discusses the experimental results. Finally, Section 6 concludes the paper and discusses future work.

## 1 Related work

Several research papers have studied automatic parallelization and how to improve dependence detection. Nevertheless, most of these studies perform parallelization guided by static data dependence analysis techniques. Also, these techniques require re-execution of the target program, and it is not generally possible to conduct parallelization automatically at runtime.

Notable static analysis work is based on the polyhedral model. The polyhedral model is a mathematical framework that represents loop nests as polyhedra whose facets are computed from the program's loop bounds. Polyhedral transformations are typically applied by a static compiler on its intermediate representation [8]. Pradelle et al. [9] proposed an automatic static parallelizer for binary code which extracts high-level information by parsing the binary code. According to this information, the system generates a C program which is parallelized via polyhedral parallelizers. Then the semantics of the original source is re-introduced and recompiled by C compiler. Therefore, this system analyzes the program statically. Furthermore, it requires re-generation of high-level program, re-compilation and re-execution.

Jimborean et al. [10] have studied dynamic and speculative polyhedral parallelization based on compiler-generated skeletons. These skeletons are applied at runtime to the original code through polyhedral transformations. Skeletons are generated at compile-time, then they are selected and instantiated at runtime. The model requires all loop bounds and memory access functions to be affine functions of global variables or outer loop induction variables. Appropriate regions are defined as Static Control Parts (SCoPs).

Yukinori et al. [11] have introduced a system to monitor binary code to check the data dependencies between memory references and dynamic loop- or call-contexts. Therefore, the system extracts the dynamic memory data-flow for the program binary code. Then, this dataflow is provided with hints to the programmers in order to re-executing the program with parallelization technique correctly.

Gupta et al. [12] have provided a static analysis technique using divide-and-conquer algorithms. This technique is studied at compile-time which benefits from the symbolic array analysis for dependence detection. The approach is applied with speculative parallelization.

Rugina et. al [13] have introduced a compiler which parallelizes recursive functions. This approach uses pointer and symbolic analysis to extract the independent recursive calls. The symbolic information provides the compiler with the main information needed to extract the procedure calls which should be executed concurrently without any violation. This approach is applied statically and generates a code to be re-executed in parallelization.

Caamano et al. [14] have proposed an automatic speculative polyhedral loop optimizer. On-the-fly polyhedral optimizations are applied on nested loops. These loops include memory access patterns and its control flow which may not computed at runtime. Apollo detects the affine expressions of the memory accesses for nested loops to be applied to speculative transformation [15]. Thus, the technique could be applied on any kind of nested loops which may contain memory accesses through pointers or indirect accesses as well as the unpredictable control. The method constructs a predictive polyhedral model to be utilized by speculative optimization. The system requires preparations to get the computed perfect version. These preparation should lead the system to re-compile. Moreover, Apollo technique is limited to the Polyhedral model.

 

AI is generally used for dependence checking in functional programming languages [2, 16, 17]. However, it is also considered for imperative languages [18]. Tzolovski et al. [19] have introduced an initial discussion on abstracting dependence. Moreover, they have explained how AI could be utilized for dependence properties such as dependence distance and iteration data dependence graphs. Also Ricci et al. [20] have proposed a static AI dependence technique to parallelize loops. They have used the interval and bisection domains to extract dependencies at compile-time. The system is implemented using PAG, an automatic analyzer [21]. PAG includes a set of codes which define the elements and the domain order during the application of the technique. Therefore, the approach requires special automatic software for implementation. Moreover, this method is applied at compile-time and it is not utilized automatically with the parallelization system.

Our work differs mainly from the related work in that we conduct analysis at runtime. This provides much accurate dependence detection. Moreover, the method is automatic and dynamic which does not need any additional directives which restricts the speculative parallelization to re-execute the code. Furthermore, our technique detects dependence correctly in the case of pointer aliasing which refers to existing of two pointers with the same value. If a pointer aliasing occurs, this may cause a dependence during the parallelized execution. Also, our system has no information about the fact that loop bounds/array indices are linear or not. This fact could be leveraged by any optimization that needs this information. Therefore, our approach does not need to build a polyhedral model. The main focus is on ranges which are determined during runtime by observing the addresses included in the hot trace. The method is able to handle the affine and semi-affine codes because of the dynamic monitoring of the addresses of the program points during runtime. Thus, our proposed technique still proves that a loop is free of dependence.

## 2 Background

In our method, we rely on conducting dynamic binary analysis (DBA) using AI to run the program using speculative parallelization. Therefore, we provide the necessary background on speculative parallelization, AI, DBA, and the Padrone binary modification framework.

### 2.1 Speculative parallelization

Nowadays, compilers mainly seek to parallelize programs relying on various parallelization techniques. Automatic parallelization techniques may be categorized into two main classes. The first one is inspector-executor scheme [22] which works by extracting an inspector loop from the original loop with some directives to lead the executor of the loop. The second one is the speculative parallelization which depends on executing the code in parallel while monitoring by a reference for avoiding violations because of data dependence [23]. In speculative systems, the analysis of data dependence is mainly studied at the case of memory access for the indices of the arrays at loops [24].

In speculative systems, the threads choose parts or all of the code from the sequential program to be parallelized. The parallel execution should preserve the semantics of the sequential run. Therefore, every speculative approach should study the data dependence precisely. The dependence violation occurs when the memory references are detected as interleaved through threads. The main dependence violations are demonstrated as follows [1]:

- Read-After-Write (RAW) dependence reads a value from memory location before the correct value is computed and written in the same location.

- Write-After-Write (WAW) dependence writes in a memory location after the last computation or calculation is stored in the same memory location. The actual final write will not be correct.

- Write-After-Read (WAR) dependence which computes, accesses and stores a new value in a memory location before the preceding read from the location itself is done.

The speculative parallelization goes through three main stages which are explained as follows [25]:

1. Defining all memory operations required for speculative execution. These operations are the main speculative data used to extract the data dependence using different techniques.

2. Preserving the speculative current state during runtime ready for any detected violation. This preservation is performed by buffering all speculative data. If there is no detected dependence, so these data are prepared to be committed.

3. If there is any monitored dependence breaking, the speculator would do some rolling-back and restarting operations for the threads sequentially.

In this paper, we study how to manage the first stage efficiently. Our system extracts the memory operations of the binary instructions in the hot execution regions of the Padrone extracted hot functions. These memory operations require a detection of the incured data dependence at runtime. Our proposed approach would be able to pause the Padrone from its original execution of the program to begin our interpreter/emulator. Therefore, our system emulates the execution to analyze the hot executing trace of the extracted function during the first number of iterations. Then, a dependence map is provided to the speculator which would run after resuming the Padrone execution. The speculator would utilize the dependence map to parallelize this hot execution region or not in the same run. Our approach could monitor the main behavior of the addresses. This monitoring would be utilized to convert the used addresses concrete domain into abstract domain. This abstract domain would be applied to extract any dependence in all types of accesses, direct or indirect. Also, the system would be applied for linear and non-linear systems.

## 2.2 Abstract interpretation

AI is a static analysis technique that is based on abstracting program semantics. The abstraction allows for fast and sound analysis. AI defines an abstract domain to represent the collective program semantics at each program point. The collective semantics are the set of all possible state values at a specific program point for all program execution paths/traces. Our analysis collects the semantics through the AI version of the code execution at binary-level. This execution guarantees monotonicity, and therefore reaching a fixpoint for the state (collective semantics for the whole program point) is guaranteed [26].

More formally, AI [5, 27] can be defined as the tuple $\langle D_a, D_c, \alpha, \gamma, I \rangle$, where $D_a$ is the abstract domain and is a complete lattice with ordering $\leq$, join operation $\cup$, and meeting operation $\cap$; with lattice bottom $\bot$ and top $\top$; the $D_c$ is the concrete domain with similar definition as $D_a$.

The functions $\alpha$ (abstraction) and $\gamma$ (concretization) define a connection between the two domains. This 'Galois' connection is computed to formalize the abstraction of each program point. The Galois connection is formalized as follows:

$$\forall i \in D_c, i \leq \gamma(\alpha(i)) \tag{1}$$

and

$$\forall j \in D_a, \alpha(\gamma(j)) \leq j \tag{2}$$

Finally, *I* defines the interpretation of each basic binary instruction, in terms of equations. Interpretation here defines the abstract collective semantic (i.e. how to compute the output of the instruction given an input). The analysis is then carried out by iterating through *I* equations, until reaching a fixpoint. The solution of these equations refer to the fixpoint at the current program point.

### 2.3 Dynamic binary analysis

Dynamic Binary Analysis (DBA) is a method to analyze the binary instructions of executing code. The main advantage of dynamic binary instrumentation is that the client program does not need any special programs to make instrumentation [28]. The analysis could be done easily, and the dynamic analysis tool is suitable for all users. Also, the code is analyzed even if the input code is dynamically generated. This is impossible with static analysis tools [29]. DBA is classified into three main popular techniques. Firstly, tracing symbolic execution would be able to follow the spread of specific sensitive data which is called taint analysis [30]. Secondly, a dynamic analysis is able to generate the control-flow for a given input binary code. Finally, the dynamic information extraction can extract runtime values at each program point [31]. In our work, the proposed system exploits the application of the last technique of extracting the dynamic information to facilitate the ability to get the runtime values.

### 2.4 Padrone framework

Padrone [7] is a framework for dynamic binary analysis and optimization. It provides the basic functionality to manipulate binary programs in memory, as well as an interface that lets clients write their own tools. Padrone relies on performance hardware counters as much as possible to identify the critical regions of a program and extract basic performance indicators (such as cycles spent, or instructions per cycle). This permits extremely lightweight detection of hot functions. Regarding optimization, the design of Padrone differs from many other tools (such as Pin, DynamoRIO). Padrone executes most code in-place, and relies on a code cache only for the modified program fragments. Padrone also executes in a different Linux process from its target: it attaches through the `ptrace` system call, interacts with the `GET/SET_REGS` action request or with direct access to the target's address space mapped to `/proc/PID/mem`, and eventually detaches.

## 3 Runtime AI approach

This section introduces the main concept of our proposed method, which is conducting AI at runtime. In particular, the code region is defined through interpreting the binary instruction, thereby revealing control-flow among instructions. As mentioned earlier, AI computes the collective semantics of the program using a set of equations. These equations consider the data-flow of the program during runtime. In our work, our system saves relevant predecessor instructions' collective semantics through a store structure. Therefore, each current instruction could retrieve its predecessors' previous values.

In the binary-level, each equation declares a program point which accumulates the semantics according to the join of all reaching definition and the corresponding instruction's operation. The program point here is defined as the arc between every two successive instructions.

We can explain briefly our main approach using the following example:

```
1 int x[1000] = {0}, y[1000] = {0}, z[1000] = {0};

2 int i = 0, j;

3 . . .

4 while(i < 1000) {

5    j = i;                        // S1

6    while(j < 500) {

7        x[j] = y[j] − z[j];       // S2

8        j++;

9    }

10   while(j < 1000) {

11       y[j] = x[j] + z[j];       // S3

12       j++;

13   }

14   i++;

15 }
```

As shown in the example, the statements *S2* and *S3* may include dependence because of *y* and *x* arrays. This dependence may be RAW or WAR. However, our approach includes the actual values, concrete values, stored in the indices *i* and *j* during actual execution. These concrete values are applied to the abstract domain as abstract intervals. The abstract intervals will contains all possible range of memory locations which may be read from or written to by *x*, *y* and *z* arrays. Also, the system computes the data-flow analysis to facilitate the definition of all collective semantics at each instruction at the same run. Therefore the method computes the abstract domain at runtime and reaches the fixpoint after small number of iterations. In the shown example, the approach can compute that index *j* will be [0, 499] in the first loop, and it will be [500, 999] in the second loop. Thence, the statements *S2* and *S3* do not contain any type of dependence.

For more clarification, the approach is explained using another example code fragment. The high-level code of the example is presented as follows:

This high-level code presents a loop which include WAR dependence at line 8. This line includes an addition operation between three arrays s[i], a[i], and b[i]. The result is stored in s [i]. Thus, s[i] is used as read and write at the same iteration at high-level. In the following code, we present the extracted equations of AI according to our proposed method.

In line 5, if there is a flag, *rare*, which is not visited during our executing and analyzing, this condition will not be analyzed. However, our system inserts some guards in the case of existing unseen code parts. If this condition is visited during speculation, there is a guard to inform the speculator that these instructions are not analyzed.

The example shows a loop that contains no loop-carried dependence, however there is a WAR dependence inside the same loop iteration. At each control-flow edge, abstract equations

```
1 int s[100] = {0}, a[100] = {0}, b[100] = {0};

2 bool rare = false;

3 int i = 0;

4 while (i < 100) {

5    if (rare) {

6        . . .

7    }

8    s[i] += a[i] + b[i];

9    i++;

10 }
```

are defined (which are points 1, 2, . . . 6). We consider the interval domain as the abstract domain.

The following code demonstrates the computed AI equations for the high-level example code:

Generally, we could compute the array abstract intervals for addresses and values. The abstract value interval would be updated as any other variable during runtime till reaching a fixpoint. Although for the addresses intervals, we need to compute the abstract addresses intervals using its index variable. Therefore, the technique extracts the abstract value interval from the index variable at runtime. Like the example, at program point 4, the `sia_r4` refers to the address for the array s using index i at point 4. The variable `sia_r4` is computed by adding the intervals of addresses of `s1` as a base address of the array and `i3` as the collective index value interval. Also, `sia_w4` is the address interval of array `s1` using index i and w is for the write operation. The variable `sia_w4` will use address interval of `s1` and value interval of `i3`. Each of the read and write operations are done in separate assembly instructions. Therefore, the method could detect any dependent pairs at runtime.

The following is a portion of assembly instructions corresponding to the equation `s[i]+ = a[i]+b[i]` (S1–3) and `i++` (S4):

This code has a number of operations that access the memory like S1, S2, and S3. The array access is inferred through the instruction format of having a fixed base address, scale, and an index register.

During runtime, there may be branches which contain variables that are affected by AI. Therefore, we need to compute the new abstract intervals that should be true inside each branch. This would be using the intersection operation between the collective semantic of the variable in the condition and the wide interval that achieves its condition. For example, *if-then* condition is less than `10`, so the widened interval would be [-inf, 9]. In our example, for the condition in the *while* loop, the widened interval would be [-inf, 99]. Furthermore, collective semantics of i are constructed, then the approach evaluates their intersection with the widened interval. Thus, we could get the abstract interval at the condition at the point 3.

The Arithmetic Logic Unit (ALU) is used to perform arithmetic and logic operations like addition, subtraction, etc. [26]. Our method applies AI on ALU operations at binary-level.

int s[100} = {0}, a[100] = 0, b[100] = {0};

1. s1 = [<s Address>,<s Address>]

   sv1 = [0, 0]

   a1 = [<a Address>,<a Address>]

   av1 = [0, 0]

   b1 = [<b Address>,<b Address>]

   bv1 = [0, 0]

**int** i = 0;

2. i2 = [0, 0]

**while**(i < 100) {

3. i3 = i2 ∪ i5 ∩ [-inf ,99]

   s[i] += a[i] + b[i];

4. //Mem read addresses

   sia_r4 = s1 + i3

   aia_r4 = a1 + i3

   bia_r4 = b1 + i3

   //Mem write address

   sia_w4 = s1 + i3

   sv4 = (sv1 ∪ sv4) + av1 + bv1

   i++;

5. i5 = i3 + [1, 1]

}

6. i6 = i2 ∪ i5 ∩ [100, inf]

---

```
mov edx, [rax*4+0x601200]      //S1
..
mov eax, [rax*4+0x6013a0]      //S2
..
mov [rax*4+0x601200], edx      //S3
add dword [rbp-0x4], 0x1        //S4
```

Therefore, every interval includes its lower bound (LB) value and upper bound (UB) value. For example, for the addition operation between I1, [LB1, UB1], and I2, [LB2, UB2], it could be performed as follows:

```
LB = LB1 + LB2
UB = UB1 + UB2
I = [LB, UB]
```

Thus, these intervals could be computed for all ALU operations using the same method.

The binary instruction is mapped into an abstract equations. Each equation includes right-hand side which represents the input abstract intervals and left-hand side representing the output abstract intervals. Furthermore, the abstract semantics are interpreted till convergence. The abstract state reserves the collective semantics for the hot trace of the extracted hot function.

The hot function which contains the hot loop may include function call(s). These function calls are not studied or handled in our method because each function call needs complex requirements which are not solved in purpose of parallelization [32]. Once our system detects the function call, then the analysis would terminate.

The program and its corresponding abstract equations are executed/evaluated dynamically according to the control-flow. If all reachable instructions are visited at least two times with no state change (fixpoint), the analysis is terminated, thereby the approach could compute the solution of the abstract equation and compute the collective semantics for each program point. The example code will be further analyzed in Section 4 showing the fixpoint collective semantics.

## 4 Our method design and implementation

The system receives a pre-compiled executable binary code. Padrone lightweight profiling computes the hot function for this binary program. Once the hot function entry is reached, Padrone pauses the execution of the program and take a private copy of the execution status (registers and memory values) to let the emulation part to start working on it. The hot function is then fed to our analysis interpreter to interpret, execute and analyze, using the abstract equations. Apparently, the analysis is performed to fetch and execute/analyze each assembly instruction in the hot execution region during runtime. The approach applies the data-flow on the fly to gather the reaching definitions and updates the abstract equations. Thus, the system can compute the abstract collective semantics at each program point. Therefore, our analyzer could compute the data dependence (through intersecting accessed addresses' intervals). Padrone framework resumes the program execution after receiving the dependence analysis through a speculative parallelization technique.

The proposed system studies the behavior of each address included in the hot trace. This hot trace consists of the hot consecutive program points that are usually executed at runtime. Also, the preserved hot trace would let the analysis compute the loop abstract intervals for bounds. The program points, which are not involved in the current hot trace, will not be analyzed. Therefore, our approach could elicit the dependent instruction pairs which are subsequent of the analyzed function hot trace.

### 4.1 Padrone lightweight profiling and hot function extraction

First, Padrone identifies *hot* functions, i.e. functions where the majority of execution time is spent. This is achieved with a very low overhead, thanks to the hardware performance monitoring unit (PMU) available in modern x86 processors. The PMU is programmed with a sampling rate, it instructs the processor to periodically register the current value of the program

counter (the address of the currently executing instruction) into a buffer. By analyzing the contents of the buffer, Padrone retrieves the most frequent addresses. For a given address, it then computes function bounds, and builds the control flow graph (CFG) [33] and applies a loop detection algorithm to build the loop forest. The overhead remains low due to the combination of two factors: (1) sampling is done in hardware, and (2) the analysis is performed in a different process from the target: the process keeps running while the application is being analyzed. The output for these steps is the extracted binary code for the hot function.

## 4.2 Interpreting and emulating

As shown in Fig 1, we have implemented an emulation of interpreter on the Padrone framework. This interpreter is used to interpret each assembly instruction as well as applying our dynamic AI technique. In the first iteration, the interpreter maps each variable value and address to the abstract domain. Moreover, the Padrone framework reads the values of the used registers and memory locations at each program point. In the consecutive iterations, the method reads the collective semantics computed in previous iterations to update the abstract domains according to the current program state. Therefore, the interpretation process is done by reading the operand and our approach could execute the operand's operation to get the result. This result would update the abstract domain for the current program point every iteration till fixpoint is reached.

Fig 1 illustrates the flow-graph for our system design. The example explained in Section 3 will be discussed in this section more clearly to explain each step in Fig 1.

Our analysis is based on the CFG for the hottest function which is excerpted by the Padrone framework. As presented in Algorithm 1, when the native execution reaches the first instruction in the extracted CFG, our analysis starts to work by reading the current status of the registers and memory. Moreover, with interpreting each instruction in the CFG, the analysis can both perform the native execution of this CFG and apply the AI technique. Our analyzer begins to update the registers/memory locations in a private memory and the AI results in hash maps. By keep going through the interpretation of the instructions flow, the approach can reach to either the exit point from the working CFG and hence exit the current analysis, or reaching a stable fixpoint from the AI analysis. Therefore, our method starts the next step to utilize the AI results to compute the dependence between instructions.

**Algorithm 1**: Dynamic abstract interpretation analysis algorithm

```
Input: CFG Hot Function Code (HFC), Program Counter (C)
Output: List of Dependent-Pair Instructions, Analyzed Region
1 C ← HFC.begin();
2 List of Dependent-Pair Instructions ← φ;
3 while C ≠ HFC.end() & AccFixpoint() = false & timeOutIteration(X) =
false do
4   FetchAndExecute(C);
5   if Var ∉ VarList then
6     Initialize VarInterval;
7     Store VarInterval in VarList;
8   else
9     for RHSVar ∈ VarList do
10      Retrieve RHSVarInterval From DFAnalysis;
11    OldInterval = VarList(C);
12    NewInterval = ApplyEquations(op);
13    if OldInterval ≠ φ then
14      CheckFixpoint(C);
15   if AccFixpoint() = true then
16     List of Dependent-Pair Instructions = DependenceCheck();
```

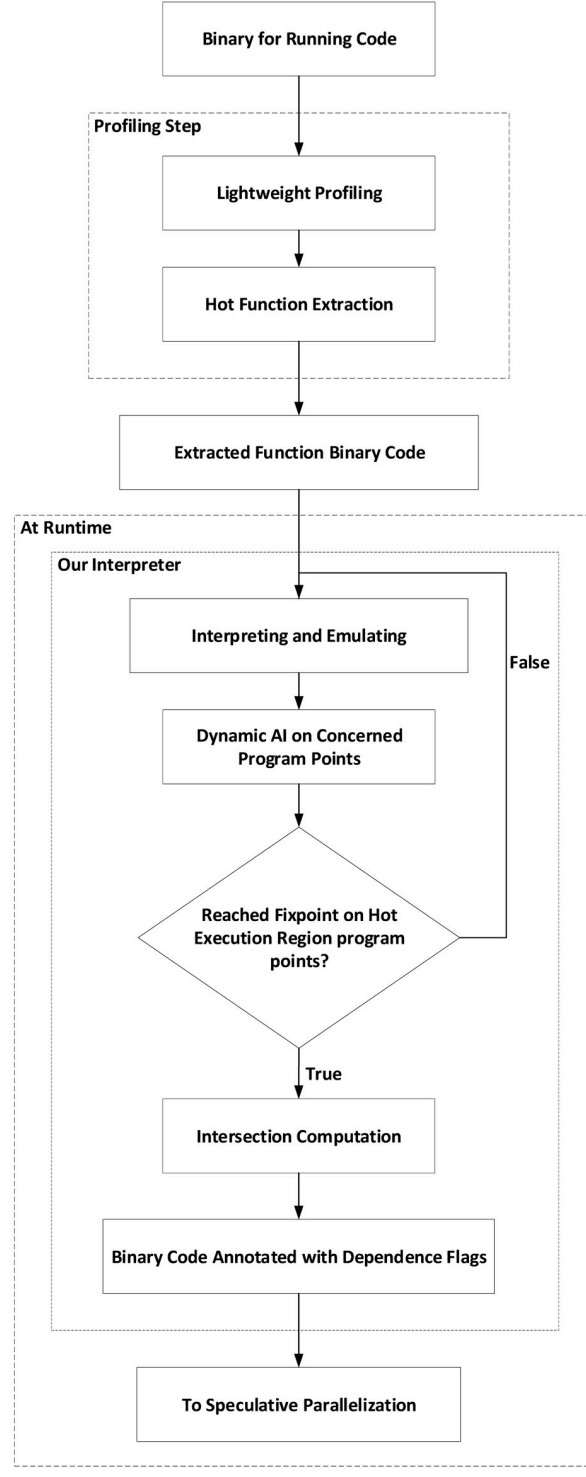

**Fig 1. System flow-graph.**

### 4.3 Dynamic AI

As shown in Algorithm 1, we define some parameters like the *HFC*, hot function code, and the program counter, *C*, for the first instruction that should run in the hot execution region of *HFC*. The counter *C* takes its initial value from the begin of *HFC*. The output of the method is the list of dependent pairs of instructions. The algorithm continuously works till reaching a legal exist point of *HFC* in the hot execution region, the *AccFixpoint* function is true, or a specified number, *X*, of iterations is reached, *timeOutIteration(X)*.

We begin the loop by fetching and executing the instruction number *C* in a native way. Then, the instruction's variables are checked whether they are included in *VarList* or not. If the variable, *Var*, is not in *VarList*, so an initial interval is stored for this *Var* as *VarInterval*. If the *Var* is found in *VarList*, the AI method could be applied on the stored interval. Each instruction's right-hand side variable, *RHSVar*, is checked whether it is included in the *VarList*. Therefore, we could retrieve the reaching definitions (and their corresponding intervals) from incremental data-flow analysis, *DFAnalysis*. Then, we need to load the current instruction's old result from the previous state as *oldInterval*. Furthermore, the operation, *op*, in *C* instruction is checked to perform the equations according to its type. The result is stored in *newInterval* which will be added to *VarList*. Each instruction checks the fixpoint of it by the equality between *oldInterval* and *newInterval*.

**4.3.1 Data-flow analysis.** To implement the AI technique, we keep all reaching definition locations for each variable in each basic block. This would save all reaching abstract intervals. By backward traversing basic blocks, our system can compute all the reaching definitions for variables. Our approach combines these data and the current abstract operation in the current program point. Therefore, it can accumulate the abstract results. This accumulation will provide accurate abstract intervals for each variable in each program point.

In the example, each operation collects the predecessors' abstract intervals. This collective semantics are merged with *union* operation to get the abstract input interval.

Loc 5: add dword X, Y

Pred(X) = {X1, X2}

Pred(Y) = {Y3, Y4}

Argument1: X1 ∪ X2

Argument2: Y3 ∪ Y4

During runtime, the method inspects the values generated from the strongly-connected basic blocks. In this example, `X1`, `X2`, `Y3`, `Y4` are all abstract states that are stored as data-flow analysis to be used in the successors' instructions at the same run. So, `X` is a result from locations `1` and `2`. Also, `Y` variable is a result from locations `3` and `4`. In location `5`, our technique unions all previous intervals for `X` and `Y`. Therefore, our interpreter could perform the abstract operation `add`. The result of this abstract operation will be stored in `X`. Thus, the system will save `X` in location `5` with a new abstract interval.

**4.3.2 Abstract operations.** The operations in binary level, for example *mov*, *add*, etc., are converted into new operations that are applied to abstract intervals. Each operation retrieves the inputs after being processed by the data-flow analysis. These inputs are the abstract

intervals. Our approach begins to save the old abstract interval for the current instruction as *OldInterval*. Then, the right-hand side variables retrieve and union the previous abstract intervals. Therefore, these intervals will be inserted into the operation as the following operations' examples. The result is saved as *CurInterval* in our list.

In the mentioned example, S4 is *add* operation instruction example:

First Iteration:

add dword [rbp-0x4], 0x1

Argument1: [0, 0]

Argument2: [1, 1]

Result: [0+1, 0+1] = [1, 1]

Second Iteration:

Argument1: [0, 0] $\cup$ [1, 1] = [0, 1]

Argument2: [1, 1]

Result: [0+1, 1+1] = [1, 2]

For memory operations such as *mov*. We present the abstract domain from S1 in our example:

mov [rax*4+0x601200], edx

Base Address: [6296064, 6296064]

Scale: [4, 4]

Index: [0, 0]

Offset: [0, 0]

[(6296064)+(0*4)+0, (6296064)+(0*4)+0]

Address Interval = [6296064, 6296064]

Here, the *mov* operation needs to compute the address of the array at the current iteration in this program point. Therefore, our system can compute the *Base* which is the base address of the array, *Scale* according to the array's datatype, current *Index*, and *Offset*. The system converts all these parameters into abstract intervals, so it can compute the current abstract state.

For memory indirect accesses, if the current executing program point generates [-inf, inf] results for the needed address for read or write, our system does not model memory locations; the whole memory is abstracted as one interval. Therefore, for this case, all reads and writes related to this instruction may not be captured.

## 4.4 Fixpoint computation

After computing the abstract intervals for the corresponding assembly instructions, our approach detects whether these intervals are fixed or not. If the all abstract intervals are fixed for two consecutive iterations in the hot execution region, then this instruction reaches fixpoint. The instruction reports with a flag when it converges. In the next iteration, our approach computes the accumulative fixpoint computation for all instructions in the entire loop. Therefore, our system should confirm that the succession of the strongly-connected basic blocks is the same for two consecutive iterations. If the whole loop instructions converge to fixpoint, the analysis stops. However, if the whole loop instructions do not report true flag for accumulative fixpoint, so our approach continues the analysis.

The hot execution region which exists in the hot function will be repeated most of the runtime, and it will be checked by comparing the abstract intervals in two sequential iterations. After three iterations, our approach begins to check the fixpoint in each instruction. If the intervals are equal, then this instruction converges and is added to fixpoint list with true value. If the intervals are different, then there are intervals that are still subject to change in their bounds during runtime. Here, we call the widening function to set the interval into its final value. The widening function checks which bound is being changed between the old and new run, then it sets this bound with its infinity value. For an upper bound (UB) change, the interval will be [LB, inf]. For a lower bound (LB) change, the interval will be [-inf, UB]. For changing in both sides, the interval will be [-inf, inf].

The list of fixpoint flags for each instruction in the hot trace will be checked. If accumulative flags value is true, then the analysis stops. If the accumulate flags value is false, then we clear the list to check the fixpoint in the next consecutive iterations. An example of fixpoint computation is as follows:

> Old Interval: [1, 2]
>
> Current Interval: [1, 3] Not Fixed
>
> Widening from [1, 3] to [1, inf]
>
> Current Interval: [1, inf]
>
> Next Iteration:
>
> add dword [rbp-0x4], 0x1
>
> Argument1: [1, inf] ∪ [3, 3] = [1, inf]
>
> Argument2: [1, 1]
>
> Result: [0+1, 1+inf] = [1, inf]
>
> Old Interval: [1, inf]
>
> Current Interval: [1, inf]
>
> Fixpoint for this instruction

As mentioned in Section 3, the index is widened to the upper bound (UB) of the indicator index interval. Also, the abstract interval of the addresses for the array will be widened, as follow:

mov [rax*4+0x601200], edx

Base Address: [6296064, 6296064]

Scale: [4, 4]

Index: [0, 99]

Offset: [0, 0]

[(6296064)+(0*4)+0, (6296064)+(99*4)+0]

Widening Address Interval = [6296064, 6296460]

## 4.5 Intersection computation

When the fixpoint is detected, the analysis stops. In this step, our system begins computing all instructions' pairs of abstract intervals intersection. If the intersection result is true, so these instructions' pairs are dependent. If there is no intersection, so these instructions' pair is independent.

In Table 1 the highlighted rows refer to the dependent pair for two instructions S1 and S3 WAR dependence. The dependent pair means that this loop contains a dependence, so the speculative parallelizer should act in a suitable way with this loop after resuming the original execution by Padrone.

## 4.6 Binary code annotated with dependence flags

After computing the intersection between instructions' pair, our approach stores this computation results into a flags map which includes the dependent pairs. This map is provided to the speculator during the same run to decide whether the loop contains dependence or not.

During runtime, our approach set guards for any unseen program points. Therefore, the speculator would handle these regions as unanalyzed parts. Hence, the speculative system could take the decision to parallelize this code part or not.

The main output of our system is a new version of binary code which is annotated with dependence flags. Furthermore, guards are provided to caution the speculative system that the current instruction or code part is not analyzed.

## 4.7 Speculative parallelization

Parallelization techniques are widely used in almost all devices (computers, smart phones, embedded systems). Speculative parallelization is one of the most challenging techniques that could exploit our proposed method to detect dependent instruction pairs. The technique of Yusuf et. al [34] (subsystem of our full design presented in Fig 1) has proposed a new

**Table 1. Part of abstract intervals for the example.**

| No. | Instruction | R/W | Abstract Interval |
|-----|-------------|-----|-------------------|
| **S1** | mov edx, [rax*4+0x601200] | R | [6296064, 6296460] |
| **S2** | mov eax, [rax*4+0x6013a0] | R | [6296480, 6296876] |
| **S3** | mov [rax*4+0x601200], edx | W | [6296064, 6296460] |
| **S4** | add dword [rbp-0x4], 0x1 | - | [1, 100] |

speculative parallelization technique which relies on our AI dependence approach. This technique utilizes the on-stack replacement to allow forking new process to enter the speculative state and kill the faulty process. The system creates a serial process which would be suspended at each checkpoint to preserve the previous work yet. Simultaneously, the parallel execution of the same program continues till reaching to the next checkpoint which is able to commit the gained work. If there is a violation occurred during runtime the speculative technique has the ability to abort the parallel execution and continue with the serial program which contains all saved work on previous checkpoints. Our system mainly provides the analysis of the dependence at the loop to be utilized by the speculative system at the current run. It does not require a new run to benefit from our method's result. Therefore, our system contains a dynamic automatic binary-level strong analysis.

In Section 5, we have studied the main metrics affected by our approach of dynamic data dependence. The experiments examines the overhead from the applied dependence analysis. Furthermore, the correctness of the system is examined using extracting the actual dependent pairs and the produced dependent pairs by our technique.

## 5 Experimental results

In this paper, we implement an emulation of interpreter in Padrone framework which relies mainly on interpreting each binary instruction by our AI technique. The method main results are generated on Intel Core i7-2670QM CPU 2.20,GHz x8 machine. This machine runs Ubuntu 14.04 LTS 64-bit Linux operating system. Furthermore, we study the main results produced from the proposed method on Polybench [35]. Polybench is called from the Polyhedral Benchmark suite. It is used to uniformize the kernels execution and monitoring. Moreover, the kernels of Polybench are used as a single file to perform the kernel instrumentation. Also, the kernels would run with non-null data initialization and live-out data dump. The loop bounds in each kernel are parametric to be utilized for general-purpose implementation. Moreover, the method impact is studied in two selected programs from SPEC CPU 2006 in terms of the overhead.

Our approach is examined by linear algebra kernels for this initial study. All used kernels include dependence in the hottest extracted function in the code by Padrone lightweight profiler. The results are demonstrated in Table 2 and Fig 2.

**Table 2. Dependence detected in kernels of polybench by our method and static AI (True and False Positives).**

| Kernel | True Positives (Our Approach) | False Positives (Our Approach) | True Positives (Static AI) | False Positives (Static AI) |
|---|---|---|---|---|
| 2mm | 1 | 0 | 1 | 2 |
| 3mm | 1 | 0 | 1 | 2 |
| bicg | 2 | 0 | 2 | 2 |
| gesummv | 1 | 0 | 1 | 1 |
| gemm | 1 | 0 | 1 | 2 |
| mvt | 1 | 0 | 1 | 2 |
| syrk | 1 | 0 | 1 | 1 |
| syr2k | 2 | 0 | 2 | 1 |
| atax | 1 | 2 | 1 | 2 |
| gemver | 1 | 0 | 1 | 1 |
| doitgen | 1 | 2 | 1 | 2 |
| trisolv | 2 | 1 | 2 | 2 |
| cholesky | 3 | 1 | 3 | 2 |
| trmm | 1 | 1 | 1 | 2 |
| symm | 3 | 1 | 3 | 2 |

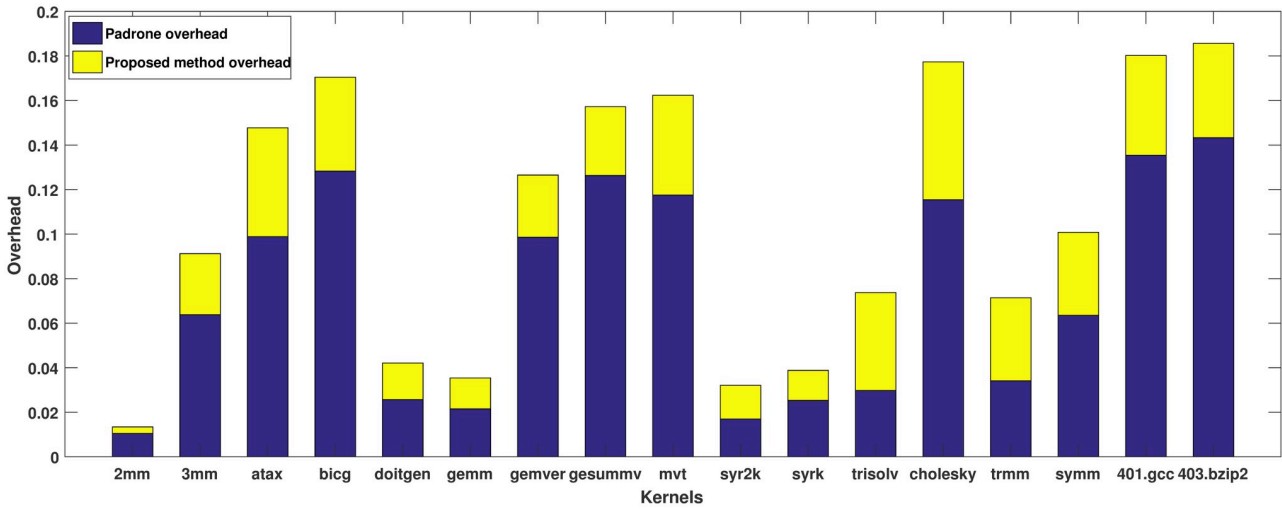

**Fig 2. Overhead between our method and original run of Polybench kernels and SPEC selected kernels which ranges from 2% to 19%.** The overhead is caused because of the analysis of large number of program instructions during runtime.

Fig 2 illustrates the overhead as a main metric for dependence detection techniques. The overhead is computed as the difference between the execution time of the kernels executed through our proposed approach and the execution time of the original running on basic kernels. The overhead is partitioned as Padrone overhead and our proposed method overhead. The Padrone overhead is ranged from about 1% to 14.3%. This overhead includes the Padrone profiling and extracting the hot function during runtime. Also, the overall overhead of 401.*gcc* and 403.*bzip*2 kernels are about 18%. On the other hand, our proposed analysis technique overhead is between 0.2% to 4.88%. Therefore, the overall overhead is mostly less than 10% which could be considered comparatively very low percentage. However, we could notice that the overhead increases on the other kernels. This increase is a result of the larger number of program instructions which are checked during runtime. This overhead will be diminished by applying the speculative parallelization in the full speculative system.

The comparison is applied between the kernels binary by native running and the kernels binary running by our AI method. The results are presented in Table 2. The first column elucidates each kernel name. Also, the hottest function exists on the inner loop of one of the main called functions. Moreover, the second column explains the true positives. This column's presented numbers are the dependent instructions pairs which are correctly dependent. The third column presents the false positives number of produced dependent instructions pairs which may not be dependent. If the false positives' values equal 0, thus the parallelizer would take the right decision to parallelize or not according to the true positives' values only. Apparently, the result of false positives is able to reflect the efficiency of any proposed approach. However, the false positives non-zero values will affect this efficiency partly. The speculator would take the decision during execution according to the preserved results of the dependence flags map. Our results prove the accuracy of our proposed approach in detecting data dependence dynamically.

All dependent pairs in Table 2 are between two instructions containing WAR dependence. The true positives, second column, are illustrated as the number of actual dependent pairs. Most of the experimented kernels contain only one dependent pair of instructions with WAR dependence which is included as a *C* equations in the for-loop. However, the kernels *trisolv*,

*bicg* and *syr2k* have two *C* statements with dependence in each one. Also, the kernels *cholesky* and *symm* include three dependencies between each pair of statements. Hence, our approach will provide the speculative system that each kernel contains dependent pair which cannot be parallelized. Moreover, false positives, third column numbers present low values in comparison with static false positives. Therefore, the missed opportunities of speculative parallelization will be lower and the accuracy will be higher than the static version. The parallelization decision would be relied on the current state as expected by our dependence analysis.

Our proposed technique have been examined in comparison with a traditional static analysis technique. This static method is based on computing the static AI intervals at compile-time as well as monitoring the definitions of registers would be used by the binary code. We have implemented a standard static AI technique using Padrone binary modification system. Padrone profiles and extracts the hot function, therefore the extracted code would be applied statically to compute data dependence. In each program point, the system checks mainly three parts. Firstly, the type of operation is checked whether it includes write, read,.. etc according to the binary instruction type *mov*, *add*, *sub*,.. etc. Secondly, the types of operators should be determined. Thirdly, the available values, if any, are inspected whether they are immediate or not. According to these three parts, the system could compute the abstract interval. Moreover, the system would use previous definitions at each program point. If there is no available values, the abstract intervals would be assumed as [-inf, inf].

The extracted hot function may include arguments. The static AI analysis is not able to detect the recent values for these arguments which affects the correctness. However, our technique would obtain the values of the arguments during runtime. These values are converted from concrete domain to abstract domain.

The impact of static AI technique is presented in terms of correctness which are shown in Table 2. the fourth and fifth columns include the true and false positives respectively. The false dependent pairs exist in all kernels with at least 1 or 2 pairs. In our method, the existence of false positives are limited to number of kernels most of cases is one pair except two kernels, *doitgen* and *atax*. Therefore, Table 2 proofs that our method correctness is higher than the applied static AI method.

For more clarification, the following example is supported to investigate the main difference in correctness between our method and static method.

The loop contains mainly three branches. First branch, *S1*, contains data dependence at *S2* and *S3*. Second branch, *S2* does not contain any data dependence. Last branch, *S6* includes data dependence for *S7* and *S8*. The branches rely mainly on the input of variables *c1* and *c2*. One of these branches will be executed during all loop iterations. Static analysis would be applied on the loop for all branches which may be executed or not. If *S4* is true, the other two branches would not be executed till the end of loop. Therefore, the loop would continue execution in parallel. However, static analysis would preserve the other two branches dependence. The speculator would resume the execution serially because of dependent program points which will never be executed. In our method, the analysis would detect that there is a branch that is executed at this run. Our dynamic analysis works for the hot trace in the hot function. Therefore, if *S4* branch is taken, the analysis would detect that there is no dependence in the current run. Subsequently, if one of the other branches is taken, our analysis would detect only the dependence in the taken branch. The other branches would not be analyzed which increase our technique's correctness.

The paper mainly achieves that we can implement a full speculative system which can detect the existence of dependence at runtime. This detection occurs at the same run with the speculative parallelization. The proposed approach can extract the dependence during the early iterations of the loops. The method will output a map of flags which will give a hint to the

```
1  int a1[1000] = {0}, a2[1000] = {0}, sum[1000] = {0};

2  int i = 0;

3  int c1, c2;

4  cin >> c1 >> c2;

5  . . .

6  while(i < 1000) {

7     if(c1 > c2) {                    //S1

8     a1[i] += a2[i];               // S2

9     sum[i] += a1[i] + a2[i];       //S3

10    }

11    else if(c1 < c2){            //S4

12    sum[i] = a1[i] + a2[i];     //S5

13    }

14    else{                        //S6

15    a2[i] += a1[i];              //S7

16    sum[i] += a1[i] + a2[i];     //S8

17    }

18    i++;

19 }
```

speculator whether the current loop contains any dependence or not. Our technique is implemented at the binary-level which would be applicable in all different architectures.

## 6 Conclusions and future work

This work proposes conducting dependence analysis at runtime for binary code using AI. During runtime, our system both interprets the seen hot execution code parts and constructs abstract equations to collect program semantics at all these parts' program arcs. Also, the system uses data-flow analysis to gather reaching definitions. Therefore, the interpretation of the code iteratively solves the equations, generating the collective semantics upon reaching fixpoint in an efficient way. The analysis is valid for a hot sub-region. Speculation recovery can be triggered upon exiting the region from unsafe points (such as not the loop exit) if used with an underlying speculative system. We have implemented the method into Padrone binary modification system using interval widening to speedup analysis. The system is fully automatic, does not require re-execution, and is applied at runtime without any special hardware/software requirements. The results show accurately the dependent pairs. Moreover, the overhead ranges from 2% to 18%.

Future work includes extending the work to consider pointer arithmetic. The work involves also studying more complex kernels with much complicated operations as some of floating point operations and vector instructions. Moreover, our system would be provided with abstracting the dependence distance between iterations. Also, we need to study the function calls handling as well as the dependence analysis during the execution of the function code. The indirections in case of [-inf, inf] would be handled as a future work to correctly get the results. Furthermore, we plan to integrate the dependence analysis work into the sub-system of speculative parallelization environment. Therefore, we would study the full system behavior and investigate the approach validation. Speculative parallelization would highly reduce the per-computed overhead and accordingly speedup the processing.

## Author Contributions

**Conceptualization:** Rasha Omar, Mostafa Abbas, Erven Rohou.

**Data curation:** Rasha Omar, Mostafa Abbas.

**Formal analysis:** Rasha Omar, Mostafa Abbas, Ahmed El-Mahdy.

**Funding acquisition:** Ahmed El-Mahdy.

**Investigation:** Rasha Omar, Mostafa Abbas, Ahmed El-Mahdy, Erven Rohou.

**Methodology:** Rasha Omar, Mostafa Abbas, Ahmed El-Mahdy, Erven Rohou.

**Project administration:** Ahmed El-Mahdy.

**Resources:** Rasha Omar, Mostafa Abbas, Ahmed El-Mahdy, Erven Rohou.

**Software:** Rasha Omar, Mostafa Abbas, Ahmed El-Mahdy, Erven Rohou.

**Supervision:** Ahmed El-Mahdy, Erven Rohou.

**Validation:** Rasha Omar, Mostafa Abbas.

**Visualization:** Rasha Omar, Mostafa Abbas.

**Writing – original draft:** Rasha Omar, Mostafa Abbas.

**Writing – review & editing:** Rasha Omar, Ahmed El-Mahdy, Erven Rohou.

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
