## [Decision Letter · Decision Letter 0]

20 Aug 2019

PONE-D-19-16831

Binary-Level Data Dependence Analysis of Hot Execution Regions Using Abstract Interpretation at Runtime

PLOS ONE

Dear Mrs. Omar,

Thank you for submitting your manuscript to PLOS ONE. After careful consideration, we feel that it has merit but does not fully meet PLOS ONE’s publication criteria as it currently stands. Therefore, we invite you to submit a revised version of the manuscript that addresses the points raised during the review process.

After a careful evaluation, we decided that the work is still incomplete and the evaluations need improving. Nevertheless, the paper has potential and can be revised to meet the PLOS ONE’s publication criteria.

We would appreciate receiving your revised manuscript by Oct 04 2019 11:59PM. To enhance the reproducibility of your results, we recommend that if applicable you deposit your laboratory protocols in protocols.io, where a protocol can be assigned its own identifier (DOI) such that it can be cited independently in the future. For instructions see: http://journals.plos.org/plosone/s/submission-guidelines#loc-laboratory-protocols

We look forward to receiving your revised manuscript.

Kind regards,

Rafael Sachetto Oliveira, Ph.D

Academic Editor

PLOS ONE

Journal Requirements:

"This research is supported by a Ph.D. scholarship from the Egyptian Ministry of Higher

Education (MoHE). Also, this work is partially funded by the PHC IMHOTEP project."

Please provide an amended Funding Statement that declares *all* the funding or sources of support received during this specific study (whether external or internal to your organization) as detailed online in our guide for authors at http://journals.plos.org/plosone/s/submit-now.  

Please state what role the funders took in the study.  If any authors received a salary from any of your funders, please state which authors and which funder. If the funders had no role, please state: "The funders had no role in study design, data collection and analysis, decision to publish, or preparation of the manuscript."

Reviewers' comments:

Reviewer's Responses to Questions

**Comments to the Author**

1. Is the manuscript technically sound, and do the data support the conclusions?

Reviewer #1: Yes

Reviewer #2: Partly

2. Has the statistical analysis been performed appropriately and rigorously? 

Reviewer #1: No

Reviewer #2: N/A

3. Have the authors made all data underlying the findings in their manuscript fully available?

Reviewer #1: Yes

Reviewer #2: Yes

4. Is the manuscript presented in an intelligible fashion and written in standard English?

Reviewer #1: Yes

Reviewer #2: Yes

5. Review Comments to the Author

Reviewer #1: This paper presents a data dependence analyzer for binary codes. An existing tool is used to manipulate the binary for extracting memory access information. Dynamic abstract interpretation is used to identify data dependencies. The paper mentions the use of such a tool to support speculative execution; however, the demonstration of its effectiveness is missing. The experimental section shows relatively low overhead.

The paper is incomplete and not organized well; however, it was not that difficult to follow. Many details are missing in this paper.

As per PLUS ONE reviewing policies, I am not evaluating how useful binary data dependence analysis is.

1. Why is the evaluation limited to Polybench? What is the rationale for evaluating the proposed work only on linear codes? Why were some of the benchmarks in Polybench excluded?

2. The authors should "clearly" specify the kind of code this framework can handle. Is this work compatible with non-linear codes? If yes, evaluate the proposed approach on SPEC benchmarks; If not why? The restriction of the approach and the implementation should be separately discussed.

3. How important is the notion of "loop iterators" and "loop bounds" for this paper? Is this work compatible with codes with indirect accesses?

4.1. Compare the scope of this work w.r.t Apollo. Apollo can support affine and semi-affine codes. Can this framework handle the same?

4.2. Can this work be used to build the Polyhedral model so that more advanced loop transformations can be applied (Apollo does something like this)? Or is the work just directed towards speculative parallelization /vectorization.

5. Does the overhead reported in Figure 2 include profiling and function extraction overheads? It is better to split the overhead of each component and redraw Figure 2 as a stacked bar chart.

6. The paper mentions speculative execution but does not present any results. Adding this data can demonstrate potential speed up.

Reviewer #2: This paper describes a new method for extracting data dependence from binary code using abstract interpretation at runtime.

This paper is in general well written and easy to understand. This helps me quickly focus on evaluating the technical work of this paper. I do have three questions that I hope the authors to address:

1. The evaluations fall short to support the claim by the author that “Our initial results the accuracy of our proposed approach in detecting data dependence dynamically”. Even though the table 2 in the paper shows that the proposed approach finds all the actual data dependency (true positives), the paper does not contain any description about false positive. At line 596, the sentence “Moreover, the second column explains the true positives.” indicates that table 2 only contains true positive numbers.

2. The experiments in this paper do not contain comparisons against any baseline. I do understand that it is likely that there is not any previous work on detecting data dependence on binary code. However, there still should be baseline to show the necessity of each component in the proposed approach. For example, I can propose a simple baseline that uses static binary level backward slicing to determine data dependence; such baseline does not incur “runtime” overhead and can be implemented with existing dataflow analysis toolkit, such as the dataflowAPI in Dyninst (github.com/dyninst/dyninst). I do not claim this baseline is better than the proposed approach. However, line 603 to 605 of the paper states that the data dependences are in “a C equation” and “two C statement”. A C equation typically correspond to several adjacent machine instructions; static analysis can extract data dependence for such cases. In other words, while the proposed approach has good potential, the evaluation does not highlight the actual effectiveness of the proposed approach.

3. In Section 4.3.2 Abstract Operations, the authors do not discuss how to handle function calls. The hot function can contain a loop in which multiple small functions are called. It is not clear to me how the proposed approach can be applied to this case.

In summary, I do believe this paper has great potential and the proposed approach looks promising, but the evaluations need improving.

6. PLOS authors have the option to publish the peer review history of their article (what does this mean?). If published, this will include your full peer review and any attached files.

Reviewer #1: No

Reviewer #2: No

---

## [Author Response · Author response to Decision Letter 0]

4 Oct 2019

All responses to comments are written in the attached file "Response to Reviewers"

---

## [Decision Letter · Decision Letter 1]

6 Mar 2020

PONE-D-19-16831R1

Binary-level data dependence analysis of hot execution regions using abstract interpretation at runtime

PLOS ONE

Dear Mrs. Omar,

Thank you for submitting your manuscript to PLOS ONE. After careful consideration, we feel that it has merit but does not fully meet PLOS ONE’s publication criteria as it currently stands. Therefore, we invite you to submit a revised version of the manuscript that addresses the points raised during the review process.

We would appreciate receiving your revised manuscript by Apr 20 2020 11:59PM. To enhance the reproducibility of your results, we recommend that if applicable you deposit your laboratory protocols in protocols.io, where a protocol can be assigned its own identifier (DOI) such that it can be cited independently in the future. For instructions see: http://journals.plos.org/plosone/s/submission-guidelines#loc-laboratory-protocols

We look forward to receiving your revised manuscript.

Kind regards,

Rafael Sachetto Oliveira, Ph.D

Academic Editor

PLOS ONE

Reviewers' comments:

Reviewer's Responses to Questions

**Comments to the Author**

1. If the authors have adequately addressed your comments raised in a previous round of review and you feel that this manuscript is now acceptable for publication, you may indicate that here to bypass the “Comments to the Author” section, enter your conflict of interest statement in the “Confidential to Editor” section, and submit your "Accept" recommendation.

Reviewer #2: (No Response)

Reviewer #3: All comments have been addressed

2. Is the manuscript technically sound, and do the data support the conclusions?

Reviewer #2: Partly

Reviewer #3: Yes

3. Has the statistical analysis been performed appropriately and rigorously? 

Reviewer #2: Yes

Reviewer #3: N/A

4. Have the authors made all data underlying the findings in their manuscript fully available?

Reviewer #2: Yes

Reviewer #3: Yes

5. Is the manuscript presented in an intelligible fashion and written in standard English?

Reviewer #2: Yes

Reviewer #3: Yes

6. Review Comments to the Author

Reviewer #2: Thanks for the authors' revision. For the responses from the authors about the three questions I raised, my comments are:

1. I do not agree with the statement "Moreover, the false positives, third column, numbers will not affect the speculative parallelization performance. The parallelization decision would be relied on the current state also as well as our dependence analysis."

If the parallelization decision relies on the dependent analysis, then false positives of the dependent analysis will lead to missed parallelization decision. Let me use an extreme example to explain. Suppose I build a dependent analysis where one instruction depends on every other instruction. This analysis will not miss any true dependency (no false negative), but cause a ton of false positive. I do not think we can make any useful parallelization decision based on such analysis with tons of false positive.

The authors provide the results showing that their approach will lead to fewer false positives compared to a base line, so false positive is a not problem for the paper. I think the false positive results improve the paper. However, I think the authors should rewrite the sentences about false positives. I disagree that "false positives will not affect the speculative parallelization performance".

2. I appreciate the authors for adding a strong baseline into the results and add a detailed examples for comparing the baseline with their approach. This definitely helps understanding the strength of the authors' approach. My honest question is in Table 2, both the authors' approach and the baseline find all the true positives in all the benchmarks used by the authors. I am not surprised by the effectiveness of the baseline approach because the authors have stated that the data dependences are in "a C equation" and "two C statements". So, the "true positives" results do not show that authors' new approach is better than the baseline. The authors' new approach is much better for false positives compared to the baseline, but the current revision seems to downplay the importance of false positives.

Therefore, I think the authors either need to find more benchmarks where the new approach finds more true positives compared to the baseline, or clearly explain the importance of false positives. Of course, the best scenario is that the authors can do both.

3. I am fine with putting handling of function calls in future work, which is indeed a difficult problem.

Reviewer #3: The paper is well written and has a good structure and organization. The Authors clearly show the limitations of current related work and the contributions of their proposal in the paper.

The authors adequately answered the questions raised by the reviewers in the revised manuscript. However, the authors need to proofread the paper for grammar in the final submission.

7. PLOS authors have the option to publish the peer review history of their article (what does this mean?). If published, this will include your full peer review and any attached files.

Reviewer #2: No

Reviewer #3: No

---

## [Author Response · Author response to Decision Letter 1]

9 Mar 2020

We have addressed all reviewers' comments in the attached file called "Responses to Reviewers"

---

## [Editor Report · Decision Letter 2]

12 Mar 2020

Binary-level data dependence analysis of hot execution regions using abstract interpretation at runtime

PONE-D-19-16831R2

Dear Dr. Omar,

We are pleased to inform you that your manuscript has been judged scientifically suitable for publication and will be formally accepted for publication once it complies with all outstanding technical requirements.

With kind regards,

Rafael Sachetto Oliveira, Ph.D

Academic Editor

PLOS ONE

Additional Editor Comments (optional):

I am pleased to confirm that your paper has been accepted for publication in Plos One.

Thank you for submitting your work to this journal.
---

## [Editor Report · Acceptance letter]

20 Mar 2020

PONE-D-19-16831R2 

Binary-level data dependence analysis of hot execution regions using abstract interpretation at runtime 

Dear Dr. Omar:

I am pleased to inform you that your manuscript has been deemed suitable for publication in PLOS ONE. Congratulations! Your manuscript is now with our production department. 

With kind regards,

on behalf of

Dr. Rafael Sachetto Oliveira 

Academic Editor

PLOS ONE